# REPAIR: REnormalizing Permuted Activations for Interpolation Repair

**Keller Jordan[1], Hanie Sedghi[2], Olga Saukh[3], Rahim Entezari[3] & Behnam Neyshabur[2]**
Hive AI[1]   Google Research[2]   TU Graz / CSH Vienna[3]
`keller@thehive.ai`, {`hsedghi, neyshabur`}`@google.com`
{`olga.saukh, rahim.entezari`}`@gmail.com`

## ABSTRACT

In this paper we look into the conjecture of Entezari *et al.* (2021) which states that if the permutation invariance of neural networks is taken into account, then there is likely no loss barrier to the linear interpolation between SGD solutions. First, we observe that neuron alignment methods alone are insufficient to establish low-barrier linear connectivity between SGD solutions due to a phenomenon we call *variance collapse*: interpolated deep networks suffer a collapse in the variance of their activations, causing poor performance. Next, we propose REPAIR (REnormalizing Permuted Activations for Interpolation Repair) which mitigates variance collapse by rescaling the preactivations of such interpolated networks. We explore the interaction between our method and the choice of normalization layer, network width, and depth, and demonstrate that using REPAIR on top of neuron alignment methods leads to 60%-100% relative barrier reduction across a wide variety of architecture families and tasks. In particular, we report a 74% barrier reduction for ResNet50 on ImageNet and 90% barrier reduction for ResNet18 on CIFAR10. Our code is available at `https://github.com/KellerJordan/REPAIR`.

## 1 INTRODUCTION

Training a neural network corresponds to optimizing a highly non-linear function by navigating a complex loss landscape with numerous minima, symmetries and saddles (Zhang et al., 2017; Keskar et al., 2017; Draxler et al., 2018; Şimşek et al., 2021). Overparameterization is one of the reasons behind the abundance of minima leading to different functions that behave similarly on the training data (Neyshabur et al., 2017; Nguyen et al., 2018; Li et al., 2018; Liu et al., 2020). Another reason is the existence of permutation and scaling invariances which lead to functionally identical minima that differ in the weight space (Brea et al., 2019; Entezari et al., 2021). Due to the relationship of the loss landscape with generalization and optimization, a large body of recent works (Li et al., 2017; Mei et al., 2018; Geiger et al., 2019; Nguyen et al., 2018; Fort et al., 2019; Şimşek et al., 2021; Juneja et al., 2022) study the loss landscape of deep neural networks with the goal of navigating the optimizer to a region with desired properties, *e.g.*, with respect to flatness around the SGD solution (Baldassi et al., 2020; Pittorino et al., 2020).

Early work conjectured the existence of a *non-linear* path of non-increasing loss between solutions found by SGD (Freeman and Bruna, 2016; Draxler et al., 2018) and empirically showed how to find it (Garipov et al., 2018; Tatro et al., 2020; Pittorino et al., 2022). Recently, Entezari et al. (2021) conjectured the existence of such a *linear* path between SGD solutions if the permutation invariance of neural networks' weight space is taken into account. That is, with high probability over SGD solutions, for each pair of trained networks A and B there exists a permutation of the hidden units in each layer of B such that the linear path between A and the permuted network B (B') is of non-increasing loss relative to the endpoints. This conjecture is important from both theoretical and empirical perspectives. Theoretically, it leads to a drastic simplification of the loss landscape, reducing the complexity obstacle for analyzing deep neural networks. Empirically, linear interpolation between neural network weights has become an important tool, having recently been used to set state of the art accuracy on ImageNet (Wortsman et al., 2022a), improve robustness of finetuned models (Wortsman et al., 2022b; Ilharco et al., 2022), build effective weight-space model ensembles (Izmailov et al., 2019; Frankle et al., 2020; Guo et al., 2022), and constructively merge models trained on separate data splits (Wang et al., 2020; Ainsworth et al., 2022). Therefore, any improvements toward reducing the obstacles to interpolation between trained models has the potential to lead to empirical progress in the above areas.

Figure 1: **REPAIR improves the performance of interpolated networks by mitigating variance collapse.** In each experiment, we interpolate between the weights of two independently trained networks whose hidden units have been aligned using the method described in Section 2.3. We then compare the interpolated network before and after applying our correction method REPAIR. **Left:** The variance of activations in interpolated networks progressively collapses. We report the average variance across each layer, normalized by that of the corresponding layer in the original endpoint networks. REPAIR is designed to correct this phenomenon. **Middle:** REPAIR reduces the barrier to linear interpolation between aligned ResNet50s independently trained on ImageNet by 74% (from 76% to 20%). **Right:** REPAIR reduces the interpolation barrier across many choices of architecture, training dataset, and normalization layer. For each architecture/dataset pair we vary the network width; larger markers indicate wider networks.

Prior and concurrent works on linear interpolation (Singh and Jaggi, 2020; Entezari et al., 2021; Ainsworth et al., 2022) have focused on improving the algorithms used to bring the hidden units of two networks into alignment, in order to reduce the barrier to interpolation between them. Singh and Jaggi (2020) develop a strong optimal transport-based method which allows linear interpolation between a pair of ResNet18 (He et al., 2016) networks such that the minimum accuracy attained along the path is 77%. This constitutes a "barrier" of 16% relative to the original endpoint networks which achieve over 93% accuracy on the CIFAR-10 test set. Entezari et al. (2021) use an approach based on simulated annealing (Zhan et al., 2016) in order to find permutations such that wide multi-layer perceptrons (MLPs) (Rosenblatt, 1958) trained on MNIST (LeCun, 1998) can be linearly interpolated with a barrier of nearly zero. Ainsworth et al. (2022) make the first demonstration of zero-barrier connectivity between wide ResNets trained on CIFAR-10 by replacing[1] the standard Batch Normalization (Ioffe and Szegedy, 2015) layers with Layer Normalization (Ba et al., 2016), and develop several novel alignment methods. Further discussion of related work can be found in Appendix A. In this paper we are interested in understanding why alignment of the endpoint networks alone has so far been insufficient to reach low-barrier linear connectivity between standard deep networks.

**Contributions** In this work we focus on understanding the source of the poor performance of standard deep networks (ResNet18, VGG11) whose weights have been linearly interpolated from between pairs of networks with aligned neurons. Our contributions are as follows:

- We find that such interpolated networks suffer from a phenomenon of *variance collapse* in which their hidden units have significantly smaller activation variance compared to the corresponding units of the original networks from which they were interpolated. We further identify and explain the reason behind this variance collapse. (Figure 1 (left) and Section 3).

- We propose REPAIR (REnormalizing Permuted Activations for Interpolation Repair), a method that corrects variance collapse by rescaling hidden units in the interpolated network such that their statistics match those of the original networks. (Section 4).

- We demonstrate that applying REPAIR to such interpolated networks leads to significant barrier reductions across a wide variety of architectures, datasets, normalization techniques, and network width/depth (Section 5 and Figure 1 (middle and right)).

## 2 PRELIMINARIES

In this section we give preliminary definitions and algorithms to be used throughout the paper.

### 2.1 LINEAR INTERPOLATION OF NEURAL NETWORKS

We consider the problem of interpolating between independently trained neural networks. That is, if we let $\theta_1, \theta_2$ be the weight vectors of two such networks, then we are interested in networks whose

---

[1]See the code release, `https://github.com/samuela/git-re-basin/blob/main/src/resnet20.py#L18`

weights are of the form $\theta_\alpha = (1 - \alpha)\theta_1 + \alpha\theta_2$ for $0 < \alpha < 1$. We refer to such networks $\theta_\alpha$ as *interpolated networks*, and to $\theta_1, \theta_2$ as the *endpoint networks*.

The *loss barrier* $B(\theta_1, \theta_2)$ between a pair of networks is defined (Entezari et al., 2021) as the maximum increase in loss along the linear path between $\theta_1$ and $\theta_2$, relative to the corresponding convex combination of the two endpoint losses. In notation:

$$B(\theta_1, \theta_2) = \sup_{\alpha \in [0,1]} \left[ \mathcal{L}((1-\alpha)\theta_1 + \alpha\theta_2) \right] - \left[ (1-\alpha)\mathcal{L}(\theta_1) + \alpha\mathcal{L}(\theta_2) \right]. \tag{1}$$

The barrier between pairs of unmodified, independent SGD solutions $\theta_1, \theta_2$ is typically large. We next discuss permutation invariance in neural networks.

## 2.2 Permutation invariance

For typical neural architectures, the neurons in each layer can be permuted without functionally changing the network; this is known as the *permutation invariance* property. In a simple feedforward network, this amounts to the observation that we can replace the $L$th weight matrix by $PW_L$ and the $(L + 1)$th weight by $W_{L+1}P^{-1}$ for a permutation matrix $P$ without changing the function represented by the network. Therefore, even if two networks $\theta_1, \theta_2$ learn functionally identical sets of neurons at each layer, it is possible for their neurons to be arbitrarily permuted or misaligned.

Entezari et al. (2021) conjecture that if permutation invariance is taken into account, then all SGD solutions for a sufficiently wide network trained on the same task become linearly mode connected. For completeness we provide the formal statement of the conjecture below.

**Conjecture 1.** *(Entezari et al., 2021) For a given neural architecture, let $\mathcal{P}$ be the set of all valid permutations of hidden units, and $P : \mathbb{R}^k \times \mathcal{P} \to \mathbb{R}^k$ be the function that applies a given permutation to a weight vector and returns the permuted version. Then for sufficiently wide networks, there exists a high-probability set of SGD solutions $\mathcal{S} \subseteq \mathbb{R}^k$ and a function $Q : \mathcal{S} \to \mathcal{P}$ such that for any $\theta_1, \theta_2 \in \mathcal{S}$, we have small interpolation barrier $B(P(\theta_1, Q(\theta_1)), \theta_2) \approx 0$.*

## 2.3 Neuron alignment algorithms

A number of works have proposed methods of finding such alignments between the hidden units of a pair of neural networks. Li et al. (2015) propose to maximize the sum of correlations between the activations of paired neurons across a batch of training data. That is, if we let $X_{l,i}^{(0)}$ and $X_{l,i}^{(1)}$ be random variables corresponding to the activations of the $i$-th hidden units of the $l$-th layer (across a batch of training data), then we optimize the permutation $P_l$ to minimize following cost:

$$\sum_i \text{corr}(X_{l,i}^{(1)}, X_{l,P_l(i)}^{(2)}). \tag{2}$$

This amounts to a linear sum assignment problem corresponding to the matrix of correlations between pairs of hidden units in the two networks which can be solved via the Hungarian algorithm (Kuhn, 1955). Recent works have proposed alternative approaches: He et al. (2018) compute a Hessian approximation to align functionally similar neurons, and Singh and Jaggi (2020) develop an optimal transport-based method of soft-alignment. Ainsworth et al. (2022) compare three methods, including one based on Li et al. (2015) and two novel approaches.

Tatro et al. (2020) also perform alignment based on minimizing Equation 2, in order to reduce the barrier to non-linear interpolation. They show using a proximal alternating minimization scheme that alignments found using this method are nearly optimal for their purposes. In this paper, we continue to use this alignment method which was originally introduced by Li et al. (2015).

We note that for networks with residual connections, care must be taken to restrict the set of permutations such that the function represented by the network does not change. In particular, the same permutation of hidden units must be applied to all layers which feed into a single residual stream.

## 3 Variance collapse in interpolated networks

We begin our study of interpolated networks by picking up where previous works left off:

- The barrier between aligned networks decreases with width, including to nearly zero for very wide MLPs trained on MNIST. The barrier also increases sharply with depth, becoming large for MLPs or simple CNNs of more than a few layers (Entezari et al., 2021).

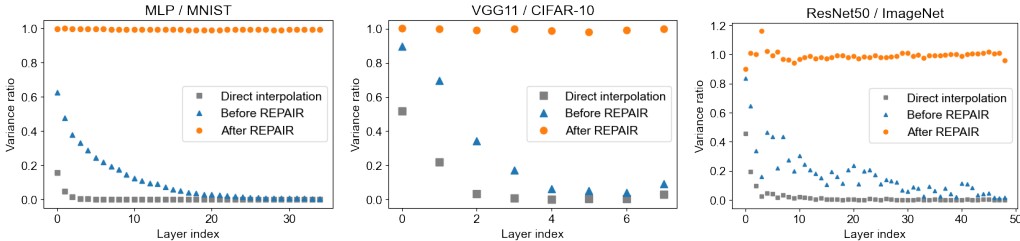

Figure 2: **Variance collapse phenomenon in averaged networks.** We find that the hidden units of weight-space averaged neural networks suffer from *variance collapse*: as we progress through the network, variance of neuron activations reduces, with neurons in deeper layers becoming nearly constant while varying the input data. *Before REPAIR* refers to networks which are interpolated from endpoint networks whose hidden units have been aligned (Li et al., 2015; Singh and Jaggi, 2020; Ainsworth et al., 2022), before our correction method REPAIR is applied. REPAIR is applied on top of this baseline in order to restore the internal statistics of averaged networks back to the level of the endpoint networks.

- Even strong optimal transport-based methods, which go far beyond alignment by allowing each neuron in network A to be matched to a weighted sum of neurons in network B, are insufficient to achieve low-barrier (below 5% test-error) connectivity between standard ResNets (Singh and Jaggi, 2020).

We focus on understanding the source of this sharp increase in barrier between aligned networks as a function of depth observed in Entezari et al. (2021), which we hypothesize to be the same phenomenon that causes a high barrier to interpolation for standard ResNets.

## 3.1 IDENTIFYING THE PROBLEM: VARIANCE COLLAPSE

What causes this rapid drop-off in the performance of interpolated networks which are deeper than a few layers? To answer this question, we investigate the internal behavior of such networks, focusing on the statistics of hidden units (Figure 2). We find that for deep MLPs, interpolated from pairs of aligned endpoint networks which have high accuracy on the MNIST test-set, hidden units undergo a *variance collapse*. That is, the variance of their activations progressively decays as we move deeper into the network, with activations in later layers becoming nearly constant. For each layer, we quantify this decay as follows. First, we measure the variance of each neuron across a batch of training data. Then we sum these variances across the neurons in the layer. Finally, if we let this sum be denoted $v_\alpha, v_1, v_2$ for the layer in the interpolated and two endpoint networks, respectively, then we return the ratio $\frac{v_\alpha}{(v_1+v_2)/2}$. We report this ratio for each layer, yielding the sequence of values in Figure 2 (left), where we see variance collapse occur in MLPs, VGG networks, and ResNets. For an example of the set of per-neuron variances in a single layer of an interpolated ResNet18, see Figure 3 (left).

We observe that variance decays to nearly zero by the final layer of an interpolated 35-layer MLP, indicating that the activations in these last layers have become nearly constant. This effect seems to be further exacerbated when directly interpolating between unaligned networks. We repeat this experiment for VGG (Simonyan and Zisserman, 2014) and ResNet50 architectures, trained on CIFAR-10 and ImageNet respectively, and find that variance by the final layers decays by more than $10\times$ (Figure 2 (middle) and Figure 2 (right)). This is a problem: if these networks have nearly constant activations in their final layers, then they will no longer even be able to differentiate between inputs.

## 3.2 WHY DOES THIS PHENOMENON OCCUR?

We argue that this phenomenon can be understood through the following statistical calculation. Consider a hidden unit or channel in the first layer of the interpolated network. Such a unit will be functionally equivalent to the linear interpolation between the respective units in the endpoint networks. That is, if we represent the unit's preactivation by $X_\alpha$ in the interpolated network, and $X_1, X_2$ in the two endpoint networks (as random variables over the input data distribution), then the equality $X_\alpha = (1-\alpha)X_1 + \alpha X_2$ holds. We will argue that the variance of $X_\alpha$ is typically reduced as compared to that of $X_1$ or $X_2$.

If the two endpoint networks are perfectly aligned and have learned the same features, then we should have $\mathrm{corr}(X_1, X_2) = 1$. But in practice, it is more typical for pairs of aligned units (whose alignment minimizes the cost function given by Equation 2) to have a correlation of $\mathrm{corr}(X_1, X_2) \approx 0.4$. When considering the midpoint interpolated network ($\alpha = 0.5$), the variance of $X_\alpha$ is given by

$$
\begin{aligned}
\mathrm{Var}(X_\alpha) &= \mathrm{Var}\left(\frac{X_1 + X_2}{2}\right) \\
&= \frac{\mathrm{Var}(X_1) + \mathrm{Var}(X_2) + 2\mathrm{Cov}(X_1, X_2)}{4} \\
&= \left(\frac{\mathrm{std}(X_1) + \mathrm{std}(X_2)}{2}\right)^2 - \frac{(1 - \mathrm{corr}(X_1, X_2))}{2}\mathrm{std}(X_1)\mathrm{std}(X_2).
\end{aligned}
$$

We typically have $\mathrm{std}(X_1) \approx \mathrm{std}(X_2)$, so that this simplifies to $\mathrm{Var}(X_\alpha) = (0.5 + 0.5 \cdot \mathrm{corr}(X_1, X_2)) \cdot \mathrm{Var}(X_1)$. With our typical value of $\mathrm{corr}(X_1, X_2) \approx 0.4$ for pairs of neurons in aligned networks, this yields $\mathrm{Var}(X_\alpha) = 0.7 \cdot \mathrm{Var}(X_1)$, a 30% reduction compared to the endpoint networks. This analysis cannot be rigorously extended to deeper layers of the interpolated network, but intuitively we expect this decay to compound with depth. This intuition matches our experiments, where we see that variance collapse becomes worse as we progress through the layers of MLP, VGG, and ResNet50 networks (Figure 2).

## 4 REPAIR

We now propose a method which mitigates variance collapse by rescaling the preactivations of hidden units in interpolated networks. We call this method REPAIR (REnormalizing Permuted Activations for Interpolation Repair).

Given an interpolated network $\theta_\alpha = (1 - \alpha) \cdot \theta_1 + \alpha \cdot \theta_2$ for some $0 < \alpha < 1$ (between aligned endpoint networks $\theta_1, \theta_2$), we select the set of hidden units or channels whose statistics we aim to correct. For example, for VGG networks we correct the outputs (preactivations) of every convolutional layer. For ResNets we correct both these convolutional outputs and the outputs of each residual block.

Our goal will be to compute affine (rescale-and-shift) coefficients for every selected channel, such that the statistics of all selected channels are corrected. Let us consider a particular channel, *e.g.*, the 45th convolutional channel of the 8th layer in an interpolated ResNet18. Similar to the analysis in the last section, let $X_1$ and $X_2$ be the values of the channel in the two endpoint networks, viewed as random variables over the input training data, and let $X_\alpha$ be the same channel in the interpolated network. Then after rescaling, we want the following two conditions to hold:

$$
\mathbb{E}[X_\alpha] = (1 - \alpha) \cdot \mathbb{E}[X_1] + \alpha \cdot \mathbb{E}[X_2], \tag{3}
$$

$$
\mathrm{std}(X_\alpha) = (1 - \alpha) \cdot \mathrm{std}(X_1) + \alpha \cdot \mathrm{std}(X_2). \tag{4}
$$

Whereas before any correction, we typically have $\mathrm{std}(X_\alpha) \ll \min(\mathrm{std}(X_1), \mathrm{std}(X_2))$ due to variance collapse. In the following, we present an algorithm which computes exact affine coefficients for each selected channel such that conditions (3), (4) become true. We also propose an approximate algorithm which avoids the use of forward passes in the interpolated network in Appendix C.

This algorithm depends first upon the computation of the statistics $\mathbb{E}[X_1]$, $\mathbb{E}[X_2]$, $\mathrm{std}(X_1)$, $\mathrm{std}(X_2)$ for each selected channel in the endpoint networks. We present a PyTorch-based approach to perform this computation in Appendix B.2. Given these values, the algorithm proceeds as follows. For each module in the interpolated network whose outputs we have identified as targets for statistical correction, we apply a wrapper, which adds a Batch Normalization layer after the wrapped module that is initially set to "train" mode. PyTorch pseudocode for this wrapper can also be found in Appendix B.2. Each such added BatchNorm layer contains affine (*i.e.*, per-channel rescale-and-shift) parameters. For a given channel incoming to the BatchNorm layer, we set the respective affine weight to $(1 - \alpha)\mathrm{std}(X_1) + \alpha\mathrm{std}(X_2)$ and the bias to $(1 - \alpha)\mathbb{E}[X_1] + \alpha\mathbb{E}[X_2]$, where $X_1$ and $X_2$ are the respective channels in the endpoint networks as in conditions (3), (4).

With the added BatchNorm layers in training mode, during execution they first renormalize their inputs to have zero mean and unit variance per channel, and then apply the given affine transformation, such that our statistical conditions (3), (4) are exactly induced with respect to each batch of input data. Next, we pass a set of training data through the network ($\sim$5,000 examples suffices) so

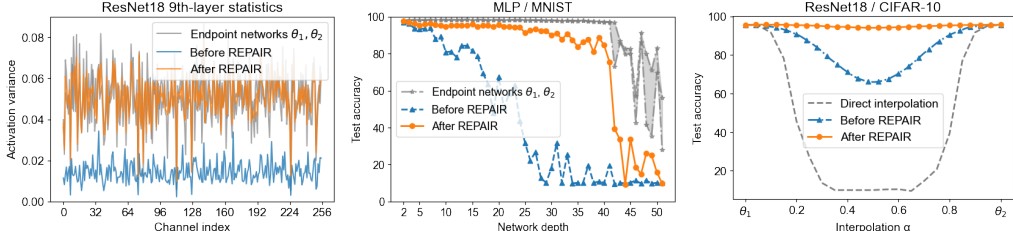

Figure 3: **REPAIR restores the internal statistics of averaged neural networks. Left:** We visualize the statistics of different channels in 9th layer of an interpolated ResNet18 on CIFAR-10. The uncorrected network undergoes variance collapse, whereas REPAIR restores the internal statistics of the network to be similar to the endpoint networks. **Middle:** Permuted interpolation without a statistical correction (before REPAIR) only performs well when limited to MLPs of a few layers (Entezari et al., 2021). REPAIR enables relatively high-performance weight-space averaging between much deeper aligned MLPs. **Right:** Using REPAIR, the barrier to interpolation between aligned ResNet18s trained on CIFAR-10 is reduced from 29% to 1.5%.

that the running mean and variance parameters of these BatchNorm layers will be accurately estimated. During this pass, any BatchNorm layers which already existed in the original network are kept frozen. Finally, we set the added BatchNorm layers to evaluation mode, so that they behave as affine layers which do not recompute statistics. At this point, the resulting network is functionally equivalent to one in which the weights of our selected set of channels have been rescaled and biases shifted. If we wish to generate a new parameter vector $\theta'_\alpha$ which is compatible with the original network architecture (*i.e.*, lacks these added BatchNorm layers), then we can perform BatchNorm layer fusion (Markuš, 2018) in which appropriate rescaling and bias-shifting values are computed from each added BatchNorm layer, and then applied to the preceding convolutional filters.

The networks resulting from this process have their internal statistics corrected, so that all selected channels satisfy conditions (3), (4). In Figure 2 and Figure 3 (left) we observe that REPAIR resolves variance collapse, bringing the statistics of the interpolated units back up to the level of the endpoint networks. In Figure 3 (left) we apply REPAIR to MLPs and in Figure 3 (right) we apply REPAIR to networks where the weights are linearly interpolated between a pair of ResNet18s whose hidden units have been brought into alignment. Before the correction, networks near the midpoint have a reduced accuracy of 66.0% on the CIFAR-10 test set, while the endpoints accuracy is 95.5%. After correction, all checkpoints along the linear path have significantly boosted accuracy, with the midpoint performing at 94.1%. In comparison, Singh and Jaggi (2020) report a linear midpoint accuracy of 77.0% using strong optimal-transport based alignment methods.

Throughout the rest of the paper, we refer to the above method as REPAIR. We note that the utility of this method is orthogonal to any improvements in the algorithm used to align the hidden units of the endpoint networks. We explore this in Appendix Figure 15, where we report barrier curves for REPAIR applied to interpolations between unaligned networks as well. In the remainder of the paper, we demonstrate the effectiveness of REPAIR across a wide range of scenarios.

## 5 EXPERIMENTAL RESULTS

In this section we report the results of the following experiments.

- We investigate the effectiveness of REPAIR for ResNets of varying width, trained on CIFAR-10 using three different normalization layers: standard BatchNorm, normalization-free using FixUp (Zhang et al., 2019), and LayerNorm. (Section 5.1)
- We apply REPAIR to the challenging scenario of interpolating between aligned ResNets trained on ImageNet. (Section 5.2)
- Finally, we perform an improved replication using REPAIR of an experiment from Ainsworth et al. (2022), in which a pair of models trained on disjoint subsets of data is constructively merged. (Section 5.3)
- We further explore the effect of width and depth across a variety of datasets. (Figure 5).

Extra figures replicating these experiments in terms of alternate metrics can be found in Appendix D.

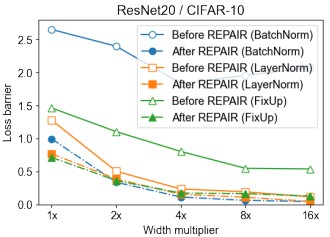 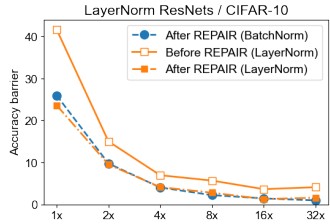 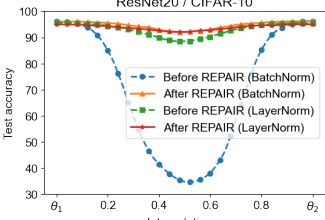

Figure 4: **Effect of normalization layer. Left:** Loss barriers with and without REPAIR for ResNet20s trained with BatchNorm, LayerNorm, and normalization-free via FixUp, varying the width multiplier from 1 to 16. **Middle:** LayerNorm networks are unique in reaching a relatively low barrier before REPAIR. **Right:** Performance curves of networks interpolated between aligned ResNet20s. Without REPAIR, the midpoint BatchNorm-based network achieves 34.7% accuracy, compared to 88.4% for LayerNorm. After REPAIR, both variants attain 92.0% accuracy.

## 5.1 NORMALIZATION LAYER AND NETWORK WIDTH

We begin by considering ResNet20s trained on CIFAR-10 using three different types of normalization layer: BatchNorm (Ioffe and Szegedy, 2015), LayerNorm (Ba et al., 2016), and normalization-free via Fixup (Zhang et al., 2019). We combine this ablation with a study on the effect of width. For each class of networks, we vary the width multiplier from $1\times$, where the final block has 64 channels, up to $16\times$ or 1024 channels. In Figure 4 we report barriers to interpolation, both with and without REPAIR. We find that REPAIR is effective for all three cases, shrinking the loss barrier to below 0.05 for BatchNorm and LayerNorm-based networks when width is scaled to $16\times$.

The LayerNorm-based ResNets are a special case that deserves comment. We first note that Layer-Norm is nonstandard in ResNets, and has worse performance when compared to BatchNorm. With our training configuration, LayerNorm-based ResNet20s achieve 92.4% CIFAR-10 test set accuracy, vs. 93.6% for the BatchNorm baseline. But these networks also have a unique property: when interpolating between aligned LayerNorm-ResNets, the midpoints already perform relatively well before REPAIR is applied, see Figure 4 (middle and right). The midpoint between our pair of $8\times$-width LayerNorm-ResNet20s attains 90.3% test-set accuracy, which is boosted to 93.2% by an application of REPAIR. In contrast, for standard BatchNorm ResNets of the same width, the interpolated network obtains only 32.3% accuracy, which is boosted to 94.4% by REPAIR.

We hypothesize that LayerNorm partially mitigates variance collapse because it applies test-time renormalization after each layer, which may prevent per-layer reductions in variance from compounding between layers. In contrast, BatchNorm is functionally just an affine layer during test-time, so that variance reductions do compound.

We claim that this observation regarding LayerNorm-based ResNets explains the contradiction between the results of Singh and Jaggi (2020) and Ainsworth et al. (2022). In the former, the authors develop a strong optimal-transport based method of aligning networks, which contains the approach of Li et al. (2015) as a special case. They show that even this method is insufficient to attain low-barrier connectivity between standard ResNets before fine-tuning; their best result is a barrier of 16% error between aligned ResNet18s trained on CIFAR-10. In contrast, Ainsworth et al. (2022) report that a variety of alignment methods, including the method of Li et al. (2015) which they call "activation matching", suffice to establish nearly zero-barrier connectivity between wide ResNets trained on CIFAR-10. How can both results be true? We claim that this contradiction is resolved by the observation that Ainsworth et al. (2022) replace standard BatchNorm layers with LayerNorm in their ResNets, as is evident in the code release[2]. In our experiments, the use of LayerNorm in ResNets uniquely allows for low-barrier linear connectivity without the requirement of a statistical correction such as REPAIR, at the cost of reduced performance on most datasets.

## 5.2 IMAGENET

Next, we explore the impact of REPAIR on the barrier to interpolation between standard ResNet models trained from scratch on ImageNet (Deng et al., 2009). We test ResNet18, ResNet50, and a

---

[2]https://github.com/samuela/git-re-basin/blob/main/src/resnet20.py#L18

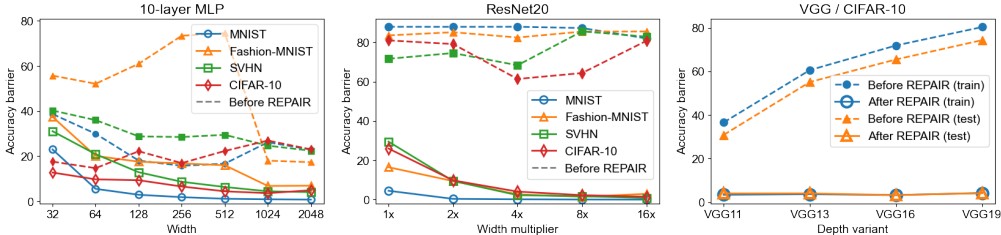

Figure 5: **Network width and depth. Left:** We investigate the effect of REPAIR on the barrier for 10-layer MLPs, trained on MNIST, FashionMNIST (Xiao et al., 2017), SVHN (Netzer et al., 2011), and CIFAR-10. In each case, the baseline (interpolation between aligned networks) is shown with dotted curve, and the solid curve refers to cases where REPAIR is applied on top of the baseline. We vary the width from 32 to 2048 hidden units per layer. **Middle:** We conduct the same experiment using ResNet20s trained on the same four datasets. We vary the width multiplier from 1 to 16, producing models whose final block ranges from 64 to 1024 channels. **Right:** We investigate the effect of REPAIR on the barrier for VGG networks trained on CIFAR-10. We vary the network depth from 11 to 19 layers, and observe that without REPAIR the barrier increases with depth, whereas with REPAIR it is close to constant in depth.

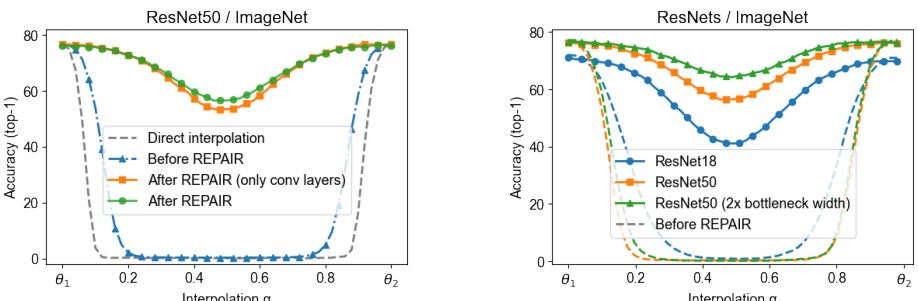

Figure 6: **REPAIR significantly reduces the barrier between ResNet50s trained on ImageNet. Left:** Without REPAIR, interpolations between two aligned, independently trained ResNet50s attain less than 1% test accuracy on ImageNet. After applying REPAIR to convolutional layer outputs, the midpoint is boosted to 53.2%. Using full REPAIR, which is also applied to the outputs of residual blocks, this is further boosted to 56.5%. **Right:** Larger and wider ResNet architectures have smaller barrier. Dashed lines indicate the baseline of interpolation between aligned networks (Singh and Jaggi, 2020; Ainsworth et al., 2022), and solid lines refer to REPAIR on top of baseline.

double-width variant of ResNet50 in Figure 6 (right). Without REPAIR, the interpolated midpoints between each aligned pair of networks perform at below 1% (top-1) accuracy on the ImageNet validation set. After REPAIR, the midpoint ResNet18 improves to 41.1%, ResNet50 to 56.5%, and double-width ResNet50 to 64.2%.

We find in Figure 6 (left) that it is important to apply REPAIR not just to all convolutional layer outputs, but also to the outputs of every residual block. For the case of ResNet50, REPAIRing these these extra channels boosted the performance of the midpoint from 53.2% to 56.5%, which is a 14% reduction in the barrier (from 23.4% to 20.1%). We note that for architectures which contain BatchNorm after convolutional layers, applying REPAIR only to convolutional layer outputs is equivalent to resetting the BatchNorm statistics. Performing such a reset on averaged networks from a single training trajectory goes back to Izmailov et al. (2018), but as far as we are aware, has not been applied to interpolations between independently trained networks until now. We provide a comparison between the ImageNet results of our work and those of Ainsworth et al. (2022) in Appendix Figure 9.

In general, the barriers for these architectures on ImageNet are still relatively high. The standard ResNet50 architecture has a final-block bottleneck width of 1024, and we measure the barrier after REPAIR to be 20.1% in terms of test error. The double-width ResNet50 variant has a final-block bottleneck width of 2048, reducing the barrier to 12.9%. In comparison, the widest ResNet20 we studied on CIFAR-10 had final-layer width of 1024, and a barrier of nearly zero. Therefore, it may be the case that for more difficult datasets, larger widths are required in order to reach low barriers.

## 5.3 SPLIT DATA TRAINING

In this section we study a setting where the two end-point networks are trained on disjoint splits of the training dataset. We aim to improve the corresponding experiment of Ainsworth et al. (2022), by using REPAIR applied to standard BatchNorm networks.

We first split the CIFAR-100 training set, consisting of 50,000 images distributed across 100 classes, into two disjoint sets of 25,000 images. The first split contains a random 80% of the images in the first 50 classes, and 20% of the second 50 classes; the second split has the proportions reversed. We then train one network on each split. The result is that the first network is more accurate on the first 50 classes, and the other more accurate on the second. Both networks underperform either their ensemble or a network trained on the full training set.

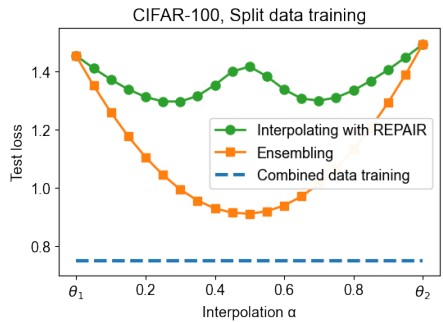

Figure 7: **Split data training.** When two networks are trained on disjoint, biased subsets of CIFAR-100, their REPAIRed interpolations outperform either endpoint with respect to the combined test set.

We next align the hidden units of these two networks, and generate a series of weight-space interpolations, applying REPAIR to each. We find that many of these interpolated networks significantly outperform either of the two endpoints in terms of loss on the full CIFAR-100 test set (Figure 7). In this sense, the two networks can be said to have been constructively merged. The best interpolated network of the corresponding experiment in Ainsworth et al. (2022) was reported to obtain a loss of 1.73 with mixing coefficient $\alpha \approx 0.3$. Using REPAIR our best interpolated network achieves a loss of 1.30, also with $\alpha = 0.3$. We attribute this improvement partially to REPAIR, and partially to the increased performance of standard ResNets compared to the LayerNorm-based variants used in Ainsworth et al. (2022). In Appendix D, we compare these results against a strong baseline.

## 6 DISCUSSION AND FUTURE WORK

In this paper we proposed REPAIR, a method of mitigating the *variance collapse* phenomenon that we identified in interpolated networks. We demonstrated that REPAIR improves the performance of interpolated networks across a wide variety of architectures and datasets. For example, we used REPAIR to reduce the barrier to permuted interpolation for standard ResNet18s trained on CIFAR-10 from 16% (Singh and Jaggi, 2020) to 1.5%, and to improve the performance of interpolated ResNet50s from below 1% to 56.5% on ImageNet. REPAIR is effective for networks trained with many choices of normalization, including BatchNorm, LayerNorm, and normalization-free.

To explain these results, we provided an analysis of the variance collapse phenomenon and how REPAIR mitigates it. We also demonstrated that LayerNorm-based ResNets are unique in attaining a relatively low barrier to aligned interpolation before the use of REPAIR, resolving the contradiction between the results of Singh and Jaggi (2020) and Ainsworth et al. (2022).

In so far as we establish low-barrier permuted interpolation for further scenarios, these results provide support for the conjecture of Entezari et al. (2021). On the other hand, to practically do so we needed to rescale the preactivations of the interpolated network, moving us out of the realm of strictly permuted linear interpolation. Our correction method REPAIR was developed as a generalization of the method of resetting BatchNorm statistics of averaged networks (Izmailov et al., 2018). Such a correction appears to be necessary in order to establish low-barrier permuted linear connectivity, at practical widths and without the use of test-time normalization layers.

On the applications side, we demonstrated that REPAIRed interpolations between two networks which were trained on disjoint, biased dataset splits are able to outperform either network from which they were interpolated. We hope that further such applied results will be possible, including improvements to weight-space ensembling, checkpoint averaging, and robust finetuning.

ACKNOWLEDGMENTS

Thanks to Luke Johnston for his insightful comments on the manuscript. This collaboration was facilitated by ML Collective.

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

APPENDIX

## A FURTHER DISCUSSION OF RELATED WORK

Entezari et al. (2021) try to find a low-barrier permutation between two SGD solutions by searching in the set of all permutations using simulated annealing. Neuron alignment methods use efficient heuristics to find a matching that maximizes a defined similarity measure given two neural networks (Li et al., 2015; Ashmore and Gashler, 2015; Singh and Jaggi, 2020; Tatro et al., 2020; Pittorino et al., 2022). This similarity measure is often based on correlation between weights or activations of the neurons in the same layer. Prior work was unable to achieve low-barrier for many standard architectures and/or challenging tasks (*e.g.*, ImageNet classification).

He et al. (2018) propose a neuron alignment method in the context of multi-task model compression. The algorithm leverages layer-wise Hessian approximation to match neurons by computing a similarity measure based on their functional difference. Singh and Jaggi (2020) propose a model fusion algorithm which leverages optimal transport to perform neuron alignment in the Wasserstein space (Agueh and Carlier, 2011), achieving a barrier of 16% for ResNet18 trained on CIFAR-10; furthermore they show that by finetuning such interpolated networks, the original performance is recovered.

Tatro et al. (2020) point out that special care must be taken for networks that contain batch normalization layers when it comes to neuron alignment. They match neurons by optimizing a curve in the weight space between two models by aligning correlations of post-activations. The running statistics are normalized by training the model for one epoch, while freezing all learnable parameters of the model. However, the crucial role of statistics reset in the context of linear interpolation of the models is not investigated. A recent work by Pittorino et al. (2022) uses LayerNorm as part of symmetry removal, so that the network behavior remains unchanged.

## B IMPLEMENTATION DETAILS

### B.1 TRAINING HYPERPARAMETERS

Table 1 summarizes the hyperparameters we used to train the neural networks which appear in this work.

- We train all networks using SGD with momentum 0.9. The weight decay and learning rates differ for each task, and are specified below.
- For our MLP trainings, we keep the hyperparameters below constant across varying widths, depths, and datasets.
- When training on MNIST and SVHN, we remove cutout and horizontal flip from the list of data augmenations used by our ResNet20 training. Otherwise, we keep the below ResNet20 hyperparameters constant across different choices of width, dataset, and normalization layer.

Table 1: Training hyperparameters

| Hyper-parameters | MLP | VGG | ResNet20 | ResNet50/ImageNet |
|---|---|---|---|---|
| Batch Size | 2000 | 500 | 500 | 512 |
| Epochs | 100 | 100 | 200 | 300 |
| Learning Rate | Linear 0.2 | Cosine 0.08 | Cosine 0.4 | Linear 0.5 |
| Weight decay | 0.0 | 0.0005 | 0.0001 | 0.0001 |
| Data augmentation | Translate | Flip/Translate | Flip/Translate+Cutout | Flip/RRC |

### B.2 PSEUDOCODE FOR PYTORCH MODULES

1. The first step of REPAIR is to measure the statistics of identified channels in the endpoint networks. There are many ways to do this, but we found that the following was efficient and had low code-complexity in a Pytorch environment. For each module in the interpolated

network whose outputs we wish to REPAIR, we wrap the corresponding modules in the endpoint networks with the following:

```python
class TrackLayer(nn.Module):
    def __init__(self, layer):
        super().__init__()
        self.layer = layer
        self.bn = nn.BatchNorm2d(len(layer.weight))
        self.bn.train()
        self.layer.eval()
    def get_stats(self):
        return (self.bn.running_mean, self.bn.running_var.sqrt())
    def forward(self, inputs):
        outputs = self.layer(inputs)
        # Apply BatchNorm so that the running mean/variance are
            updated; discard the output.
        self.bn(outputs)
        return outputs
```

It then suffices to pass a small set of training data ($\sim$5,000 examples) through the tracked endpoint networks. At this point, the statistics of each tracked module's output can be retrieved from the wrapping TrackLayer.

2. Next, we wrap each module in the interpolated network that we wish to REPAIR with the following:

```python
class ResetLayer(nn.Module):
    def __init__(self, layer):
        super().__init__()
        self.layer = layer
        self.bn = nn.BatchNorm2d(len(layer.weight))
    def set_stats(self, goal_mean, goal_std):
        self.bn.bias.data = goal_mean
        self.bn.weight.data = goal_std
    def forward(self, x):
        return self.bn(self.layer(x))
```

The next step is to iterate over each triple of corresponding (TrackLayer, TrackLayer, ResetLayer) modules coming from the two endpoint networks and the interpolated network. For each triple, we set the statistics of the ResetLayer to be the interpolation of the statistics from the two TrackLayers. Working and minimal code to accomplish this can be found in our code release.

## C CLOSED-FORM APPROXIMATE VARIANT OF REPAIR

We present an alternative correction algorithm which generates channel-wise affine coefficients using a closed-form approximation, eliminating the need for forward passes in the interpolated network. This alternative is effective for MLP networks, but we found that it did not work well for deep convolutional architectures. In Figure 8 we compare the affine coefficients produced by this variant with those of the primary method in Section 4, finding substantial correlation.

The variant proceeds as follows. Consider a hidden unit in the first layer of the interpolated network. As in Section 4, let $X_\alpha$ represent the unit in the interpolated network, and $X_1, X_2$ the same unit in the two endpoint networks, respectively. Condition (3)

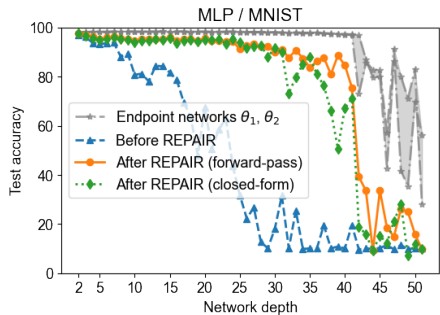

Figure 8: **Closed-form variant of REPAIR.** A closed-form, approximate variant of REPAIR performs similarly to the full algorithm for the purpose of correcting deep MLPs which were trained on MNIST.

will already be satisfied for this unit by virtue of the equation $X_\alpha = (1-\alpha) \cdot X_1 + \alpha \cdot X_2$ (this holds only because the unit is in the first layer). Given the values $\text{Var}(X_1)$, $\text{Var}(X_2)$, and $\text{Cov}(X_1, X_2)$, which can be computed only using forward-passes in the endpoint networks, it is possible to compute the variance of $X_\alpha$ exactly according to the formula

$$\text{Var}(X_\alpha) = (1-\alpha)^2\text{Var}(X_1) + \alpha^2\text{Var}(X_2) + 2\alpha(1-\alpha)\text{Cov}(X_1, X_2). \tag{5}$$

Therefore, to satisfy condition (4) for this unit, the rescaling coefficient $\beta$ must be

$$\beta = \frac{(1-\alpha) \cdot \text{std}(X_1) + \alpha \cdot \text{std}(X_2)}{\sqrt{(1-\alpha)^2\text{Var}(X_1) + \alpha^2\text{Var}(X_2) + 2\alpha(1-\alpha)\text{Cov}(X_1, X_2)}},$$

which is simply the desired standard deviation divided by the standard deviation of $X_\alpha$. For each unit in the first layer, this factor is exactly correct in order to obtain the desired statistics. For deeper layers, this factor is an approximation which we empirically test.

In Figure 3 (right), we apply this rescaling to every hidden unit of MLPs of depth between 2 and 50 hidden layers, which are linearly interpolated ($\alpha = 0.5$) between aligned endpoints networks trained on MNIST. We find that this rescaling significantly improves the performance of such interpolated networks. In particular, we obtain interpolated checkpoints of up to 27 layers that achieve over 90% accuracy, whereas without a correction, we hit this limit after only 6 layers. For this case, the approximate rescaling performs similarly to the full exact method presented in Section 4.

## D  ADDITIONAL PLOTS

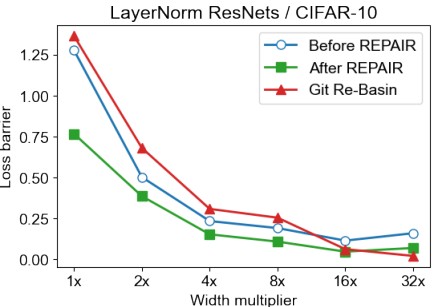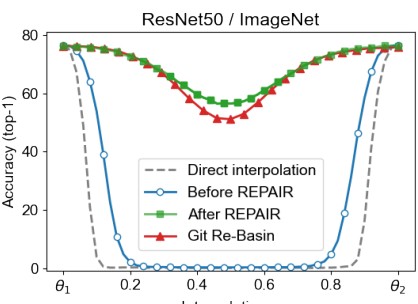

Figure 9: **Comparison to Ainsworth et al. (2022). Left:** We compare barrier values of LayerNorm-based ResNets, for our baseline and REPAIR, to the barriers reported in Ainsworth et al. (2022). We find that for networks of width up to $8\times$, our baseline barrier values are comparable, indicating that our baseline alignment method (Li et al., 2015) performs similarly to those explored in Ainsworth et al. (2022). With REPAIR, our barrier values are lower. For the case of $32\times$-width networks, our barrier value is higher than that reported in Ainsworth et al. (2022). **Right:** We compare our results on ImageNet using REPAIR with those of Ainsworth et al. (2022). We note that in their ImageNet experiments, Ainsworth et al. (2022) do reset the BatchNorm statistics in their interpolated ResNet50s based on our suggestion to do so. The extra barrier reduction in our results can therefore be attributed to the fact that REPAIR also statistically corrects the outputs of each residual block.

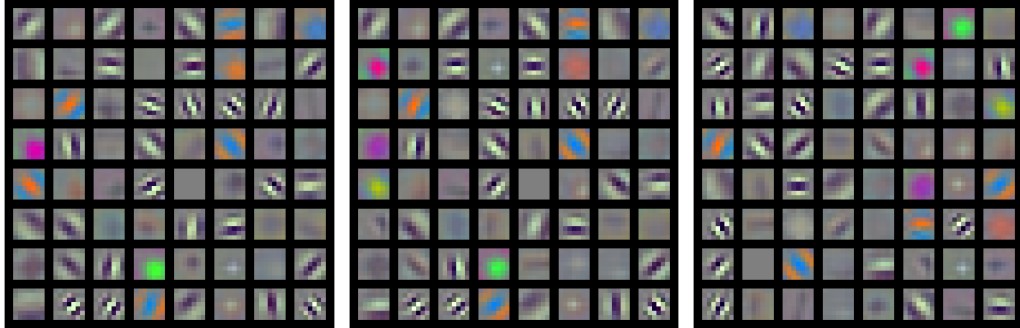

Figure 10: **Aligned convolutional filters for the first layer in ResNet50.** We train two ResNet50s on ImageNet independently, calling these models A and B. **Left:** The first-layer convolutional filters of model A. **Middle:** The filters of model B, having been permuted in order to maximize the total activation-wise correlation to the corresponding filters of model A. Some paired filters appear nearly identical, while for others there was no close match. **Right:** The filters of model B in their original positions.

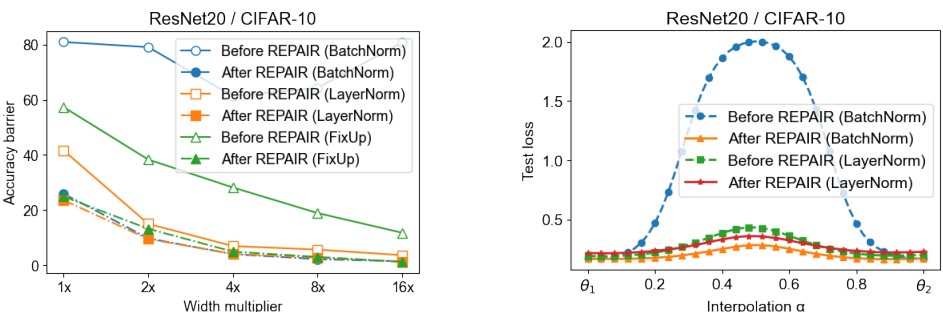

Figure 11: **Effect of normalization layer. Left:** We replicate Figure 4 (left) with barriers measured in terms of test accuracy. **Right:** We replicate Figure 4 (right) with barriers measured in terms of test loss.

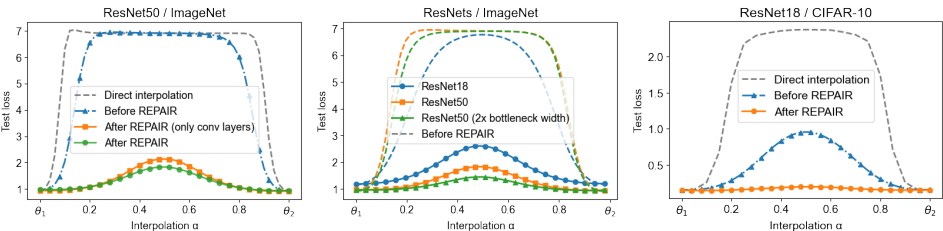

Figure 12: **Barrier curves in terms of loss.** We report the performance of interpolated ResNets in terms of test loss on ImageNet and CIFAR-10. The corresponding figures in terms of accuracy are Figure 6 and Figure 3 (middle).

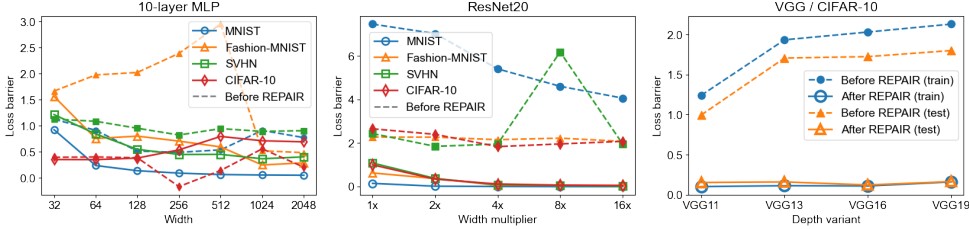

Figure 13: **Network width and depth.** We replicate Figure 5 with barriers measured in terms of test loss instead of test accuracy.

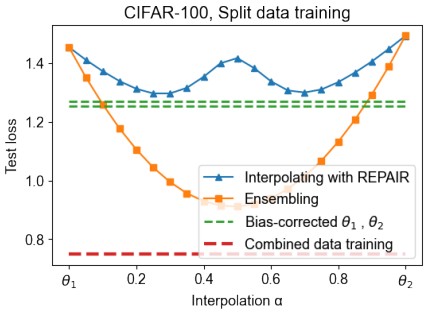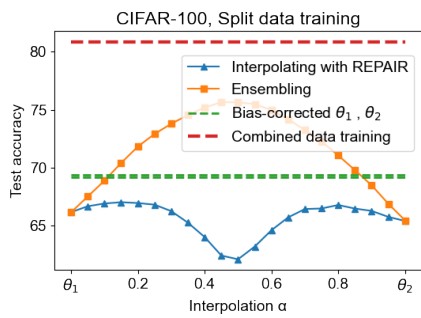

Figure 14: **CIFAR-100 split data experiment against a baseline.** Two networks are independently trained on disjoint, biased subsets of the CIFAR-100 training set. With respect to performance on the full CIFAR-100 test set, we report the performance of ensembling/mixing the logit outputs of the two models (orange), and the performance of interpolations in weight-space between two aligned models (blue), with REPAIR applied to every interpolated network. As baselines, we report the performance of a single model which was trained on the full CIFAR-100 training set (red), and the performance of the two endpoint networks, after a distribution-shift correction has been applied (green). We find that there exist REPAIRed interpolated checkpoints which outperform either endpoint network in terms of both test loss and accuracy, showing that constructive merging of models is possible. The distribution-shift correction (green line) is as follows. Consider an endpoint network A; it was trained with a training set that was biased 4:1 towards the first 50 classes, so that on the test set, network A over-predicts these first 50 and under-predicts the second 50 classes, leading to increased loss. Our baseline is then to scale down the first 50 logits of network A such that its predictions on the test set become balanced over the 100 classes. We find that this baseline, applied separately to either endpoint network (thus two green lines), outperforms the best gain that can be obtained from constructive merging of the two models via REPAIRed interpolation.

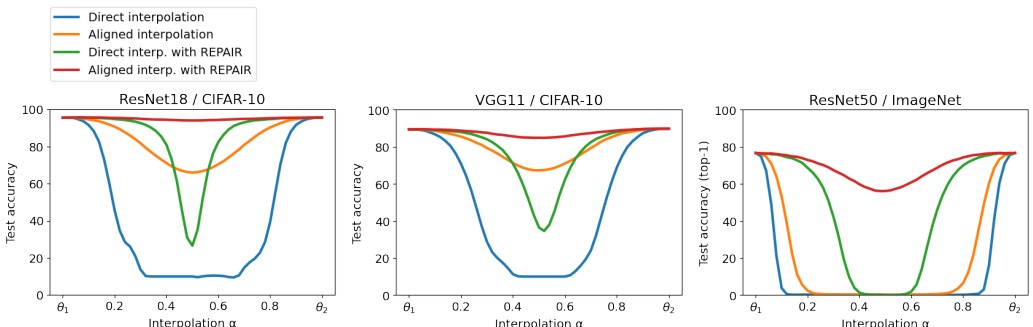

Figure 15: **REPAIR without alignment.** In the preceding experiments, we have always first aligned the neurons of the two endpoint networks before interpolating. In this figure we also report accuracy curves for interpolations between the original unaligned endpoint networks, both with and without REPAIR. We note that in the case of ResNet18, we apply REPAIR to convolutional outputs, which is exactly equivalent to resetting BatchNorm statistics. For VGG11, the architecture does not contain normalization layers, and REPAIR is applied to convolutional outputs. For ResNet50, we apply REPAIR to both convolutional outputs and residual block outputs, as described in Section 5.2. It appears that REPAIR is effective towards the endpoints, and neuron-alignment becomes essential near the midpoint. Both methods are necessary in order to have a high-performing midpoint.

