# OpenReview forum: "REPAIR: REnormalizing Permuted Activations for Interpolation Repair"
_ICLR.cc/2023/Conference — ICLR 2023 poster_

### Official Review · Reviewer_SpdK · 2022-10-23

**Confidence:** 3
**Correctness:** 4
**Technical Novelty And Significance:** 3
**Empirical Novelty And Significance:** 3
**Recommendation:** 6

**Clarity, Quality, Novelty And Reproducibility:**

Random Notes:

It would be great if the paper were readable without prior knowledge of Entezari’s 2021 paper.

3 paragraph, introduction: please define ‘loss barrier’ when you first use the term

2.2. Please give a more detailed explanation of ‘unit-rescaling’, ‘positive homogeneity of ReLU’ and why SGD helps here.

Fig 2: I don’t find some of the filters from the middle again on the right, are these the right plots? Also, there seem to be some obvious mismatches, which one could change by hand. I know the best matching may not be interpretable, just wondering if there is a bug here?

Also, Fig2, I am wondering if for a CNN having a filter which is slightly shifted (e.g. phase shifted wavelet) may give a low correlation in filter shapes, but almost the same pattern of activations in the convolutional feature map? I.e., maybe there is some translation-invariant way of matching convolutional filters?

Which algorithm are you using for linear assignment? Please provide more technical details.

Just above eq. 2 there is a superfluous ‘the’

End of 2.3. How are the weights interpolated and permuted exactly? Please give more detail.

I am little confused about the ordering of tested models and datasets in different sections. Like, for instance, starting section 3, the standard choice would have seemed like LeNet on MNIST and ResNet on CIFAR10. Also, why are models split into different sections at all, like section 5.3??

Section 4, second paragraph: WHY? Can you give any motivation for this? I guess this relates to my general sense when reading the paper: It would be great to learn something about neural networks, rather than just observing that some statistic changed, fixing it with a hack and claiming victory because performance increased.

Here is a proposal for a little more understanding of what is happening: Take a 2D slice of the loss landscape (like in other model averaging papers, e.g., the ones about model soups) and visualise the different interpolation schemes: 1) regular interpolation should be a straight line; 2) adding the constraint that the mean and variance remain the same for the units should a slight deviation – possibly through regions with lower loss? This is just a first proposal for an experiment to dig deeper, build more intuition and try to actually understand what is going on.

Fig 6: too small, and the different losses seem redundant, maybe just pick one an put others in appendix?

Fig 6: what is this? Only permuted or also variance adjusted? Add clarification.

End of 5.2: OK, I appreciate that you add Ainsworth et al. as concurrent work. However, now you are comparing against them. If you do this, then please do it properly. Include it in your plots. Show if the variance trick also improves upon the better permutation scheme from their paper etc…

Fig 8: What is this? Top1/5 accuracy? Why are those two papers cited, is it their interpolation method?

5.4.: ‘have’ +a ‘maximum barrier’

What exactly is the statement you are trying to make with Fig. 9? Surely not a causal relationship? Also, I am not sure how much of an explanation that offers at the end of section 5.

**Strength And Weaknesses:**

Strengths: The paper is very clear and well written. It is easy to follow and presents a concise observation leading to a simple modification to recover the performance in interpolated models. The experiments are straightforward and clear. They also do a good job in relating to existing research on this subject.

Weaknesses: It would be great if some more theoretical analysis were provided in an attempt to understand why variances collapse, how their modifications change the standard interpolation. While it is great to make a simple observation, propose a fix and demonstrate that it increases performance; one is left wondering, after reading the paper, what exactly we have learned from this scientific study.

**Summary Of The Paper:**

I thank the authors for their responses. I am especially happy about the added section 3.2. that tries to provide some theoretical reasons for the observed phenomenon of variance collapse. I am raising my score to 6.

The authors in this paper observe that interpolating between neural network weights decreases the variance of activations. By counteracting this phenomenon they show that much of the performance drop in interpolated models can be recovered. They show this across multiple architectures, normalisations and datasets.

**Summary Of The Review:**

The paper is clear, simple and proposes an effective modification to interpolation variance collapse. There is some concurrent work, to which it would be great to provide a more detailed comparison. Moreover, it would be great if the paper would go beyond noticing that some statistics changed, modifying this and obtaining better performance. Ideally, after reading this paper, we would have a slight advance (even preliminary, anecdotal) about why variance collapses in interpolated models, how the modification changes the interpolation path and why this works.

---

> ### Author Response · Authors · 2022-11-15
> **Response to Reviewer SpdK**
>
> We thank the reviewer for their valuable feedback. We appreciate their recognition of the clarity of our experiments and presentation. We made changes to the paper according to the reviewer’s comments, and hope that the reviewer will increase their score if they find their comments are addressed.
>
> ##  Improved theoretical analysis
> Here we address the reviewer’s main concern regarding the lack of theoretical analysis in the work. In section 3.2, we added a theoretical justification as to *why* the variance collapse phenomenon occurs. In section 4 we added more details regarding how the proposed correction method generates the exact affine coefficients for each hidden unit.
>
> We are the first to determine the required empirical correction that must be applied to realize linear connectivity between realistic networks (conjectured in Entezari et al). This correction may appear to be a hack, but we show that it is necessary to mitigate variance collapse, a general phenomenon that can be rigorously understood for the first layer of interpolated networks.
>
> ** We address the reviewer’s concerns in more detail as follows. **
>
>     the need for prior knowledge of Entezari et al.
> We updated the draft so that no prior knowledge of Entezari et al is required.
>
>     the definition of the loss barrier in paragraph 3
> In the updated draft, we introduce the “barrier” to interpolation between two networks as part of an example and visualization in the Introduction: the minimum accuracy attained along the path is 77%. This constitutes a “barrier” of 16% relative to the original endpoint networks which achieve over 93%
>
>     a detailed explanation of ‘unit-rescaling’, ‘positive homogeneity of ReLU’
> Thanks to the reviewer's comment we decided to remove the paragraph since we also believe that it may confuse the reader.
>
>     the filters displayed in Figure 2
> We appreciate the reviewer pointing out that the filters did not, in fact, match. This led us to discovering a bug in our plot in which we had been reporting the filters from three different networks, rather than two. The figure has now been moved to the appendix, and the bug has been corrected.
>
>  Our alignment algorithm does match figures based on correlation between their activation patterns rather than their weights. Therefore, the concern of the reviewer that two filters may have differing filters (perhaps where one is shifted), but the same activations does not apply to our method.
>
>     the algorithm used for linear assignment
> We use Hungarian algorithm, which is now properly cited in Section 2.3.
>
>     how the weights are interpolated and permuted
> We give a precise description of the algorithm to find permutations in Section 2.3, which is the same algorithm proposed in Li et al., 2015.
> We describe interpolation between weights in Section 2.1.
>
>     the ordering of tested models and datasets
> We agree with the reviewer that the previous version had issues with results ordering and redundancy. We reorganized our experiments to avoid any confusion.
>
>     Why the variance collapse phenomenon occurs
> We added a theoretical analysis of the variance collapse phenomenon in the first layer of interpolated networks in Section 3.2 that rigorously demonstrate why this phenomenon occurs. For our correction method, we rescale filters such that the statistics of the corresponding unit become close to those of the original networks. We describe in Section 4 why our rescaling accomplishes the necessary statistical reset.
>
>     redundancy in the figures
> We removed all redundant figures, and introduced new experiments, in particular in the context of federated learning.
>
>     how Figure 6 was generated
> Figure 6 compares ResNets before and after the application of the statistical correction. The corresponding figure in the updated draft is Figure 4.
>
>     Ainsworth et al.
> Regarding comparison to Ainsworth et al. we note that in the updated draft, we improved the application of our method to the challenging scenario of interpolation between ResNet50s on ImageNet. Our barrier has been reduced to 19.7% in terms of test accuracy, which is a 20% improvement compared to Ainsworth et al. In Appendix Figure 8, we compare our results to Ainsworth et al. Regarding the improvement of our correction on top of neuron alignment methods, in Figure 4 of the updated draft, we show that except for the case of LayerNorm-based ResNets (which are used in Ainsworth et al.), a statistical correction is essential for the purpose of low-barrier connectivity between ResNets.
>
>     the metrics used in our ImageNet figures
> Our ImageNet figures (Fig 5) are now correctly labeled as being measured in terms of Top-1 accuracy.
>
>     the purpose of Figure 9
> We moved our discussion of the relationship between the barrier size and the test error to Section 5.2, in which we make a more cogent case that networks of the same width have a much larger barrier when trained on ImageNet, in contrast to the easier task of CIFAR-10.

---

### Official Review · Reviewer_zoWW · 2022-10-24

**Confidence:** 3
**Clarity, Quality, Novelty And Reproducibility:** Paper is clear, lacks a bit of novelt…
**Correctness:** 3
**Technical Novelty And Significance:** 2
**Empirical Novelty And Significance:** 3
**Recommendation:** 6

**Strength And Weaknesses:**

The paper has following strengths:

1. It is interesting to see the variance collapse phenomenon on the interpolated networks.

2. Barrier reduction for CIFAR-10 down to 1.5% is significantly better than existing techniques.

3. The paper is well-written and nicely explained. Related work is cited and discussed very well.


The paper has following weaknesses:

1. The correction method presented by the authors is trivial. It is well-known that tuning batchnorm can significantly improve accuracy (sometimes even in completely unsupervised, domain adaptation settings, e.g., see: TENT: https://openreview.net/forum?id=uXl3bZLkr3c). Hence, it is no surprise that fixing batch norm statistics would help in this case. While the proposed method achieves great results on CIFAR-10, we can see that on ImageNet (despite the proposed method being the current best), the gap in interpolation accuracy is very significant (making this method not too useful on large-scale datasets). I was wondering if instead of just initializing affine parameters and re-computing the running mean/variance statistics, the authors were to finetune (or learn) the BN stats similar to TENT above, would we see significant reduction in the ImageNet barrier (compared to their current results)? Having better numbers for large scale datasets like ImageNet would indeed make this work useful.

2. In the final sentence, the authors state “a better understanding of linear interpolation would result in improvements in applications like weight-space ensembling, checkpoint averaging, robust finetuning, etc.”. I think the current paper lacks the use of proposed ideas (e.g., fixing variance collapse, etc.) within a real application framework. It would make the paper significantly stronger if the authors could pick a couple of the above applications and show that fixing this subtle issue significantly improves the current solutions in those domains.

3. There are many figures in the paper that do not contribute much to other results. Example, training loss barrier, test loss barrier, etc. (Fig. 6ab, Fig. 7a), do not provide any new intuitive information over the test accuracy barrier results. I suggest to remove the redundant information and include more results on downstream applications and other experiments suggested above (e.g., TENT on linear interpolations, etc.).

4. Finally, I am a bit unclear on what happens when the model already has BNs (e.g., ResNets). Do you add a new BN layer in your correction method (even if a BN layer is already present)? Page 6, “We present the following practical…” paragraph seems to suggest that you add a new BN layer. If you do add an extra BN layer: When you do interpolation, wouldn’t you already set the affine parameters (of existing BN) to the respective interpolated means and interpolated variances from parent networks (similar to all other weights in the network)? In that case, I am not sure how the new BN layer adds anything new (other than recomputing the running mean and running variance statistics with training data, which is similar to other methods like TENT that also tune the BN layer (but they also do some more learning on the BN layers)). Also, when you do the permute invariance methods, do you consider the BN layers (for somehow permuting the affine parameters, running mean, running variances)?


**Summary Of The Paper:**

The paper studies the linear mode connectivity problem empirically and uncovers a new phenomenon called variance collapse (where activation variances of interpolated networks between two parent networks reduces significantly). Test accuracy barrier reduction is shown successfully for the interpolated networks (for CIFAR-10, barrier goes down to about 1.5% gap in test accuracy between parent models and the interpolated network).

**Summary Of The Review:**

The paper can be significantly stronger if it actually shows improved results on downstream applications and manages to reduce the ImageNet barrier further with more advanced BN tuning methods.

==== UPDATE AFTER REBUTTAL ====

I have read the author response and other reviews. I am raising my score to 6 since the authors have reasonably addressed my main concerns.

---

> ### Author Response · Authors · 2022-11-15
> **Response to Reviewer zoWW**
>
> We thank the reviewer for their constructive feedback, which was a significant driver for us to update the paper draft. We appreciate the reviewer’s recognition that our method is better than any existing technique for interpolating between standard ResNets on CIFAR-10. We believe that our updated draft addresses many of the reviewer’s comments, and hope that the reviewer will increase their score, should they find that the weaknesses they pointed out have been satisfactorily addressed.
>
> ## Novelty of our method
> Regarding the reviewer’s main concern, the novelty of our correction method relative to resetting BatchNorm statistics is two-fold: (1) Our method applies equally well to networks that do not contain BatchNorm layers, including LayerNorm-based and normalization-free networks. (2) We are able for the first time to identify exactly why resetting statistics works. We find that a reset is necessary to overcome the problem of variance collapse, and our updated draft provides an analysis of why variance collapse occurs. In our updated draft we have also improved our ImageNet results by more than 50%.
>
> **We now address the reviewer’s concerns in more detail.**
>
>     Continued discussion of novelty
> The reviewer notes that “better numbers for large-scale datasets like ImageNet would indeed make this work useful.”
>
> In addition to the discussion on novelty above, we are happy to report improved numbers for our ImageNet experiments in the updated draft (Section 5.2). The size of our barrier for ResNet50 trained on ImageNet has been decreased by more than half from ~45% to 20%. One of the factors behind this improvement is that we have now extended our method to correct the statistics of the outputs of residual blocks, which yielded an extra 14% improvement on top of just correcting the BatchNorm statistics. We note that these residual outputs do not feed into the BatchNorm layers and therefore cannot be corrected with a BatchNorm reset alone.
>
> Our method also succeeds in reducing the barrier by more than 90% for VGG networks that do not contain normalization layers. As far as we know, this is not attainable by any other extant approach.
>
>     TENT
> This paper presents a method that increases out-of-distribution performance by optimizing the BatchNorm-affine parameters of a model with respect to an entropy loss on out-of-distribution data, we have the following response. First, this method requires training, i.e. backward passes against an objective and potentially multiple epochs of data. In comparison, our method only uses forward passes and a small subset of data. Regardless, we agree with the reviewer that using TENT to increase the performance of interpolated networks is an interesting direction for future research.
>
>     a real application framework
> In our updated draft we present an application of our method to a federated learning context. We train two standard ResNets on disjoint and biased splits of the CIFAR-100 dataset. We then show that using our statistical correction it is possible to effectively merge the two such that the merged network outperforms both of the original networks. This is promising from the standpoint of federated training across disjoint datasets which cannot be shared. We note that this experiment is inspired by [2].
>
>     redundant figures
> The reviewer notes that “There are many figures in the paper that do not contribute much to other results.” We agree with the reviewer and have removed redundant figures from the updated draft, replacing them with many new and more relevant experiments.
>
>     what happens when the model already has BatchNorm layers
> For the case of ResNet50, we agree with the reviewer that it would be redundant to add extra BatchNorm layers after convolutional layers, which are already followed by a BN. What is not redundant are the BatchNorm layers which are added after the outputs of each residual block. This is an addition in the updated draft which we find improves our results on ImageNet by a further 14%. This is currently the best result reported in the literature, being better than Git Re-Basin [2]. We also note that all of the added BatchNorm layers are not redundant in case of networks that either lack normalization layers or use LayerNorm.
>
>     the treatment of BatchNorm layers during permutation
> The reviewer asks “when you do the permute invariance methods, do you consider the BN layers (for somehow permuting the affine parameters, running mean, running variances)?”
> All parameters of each BatchNorm layer in the network are permuted along with the channels of the preceding convolutional layer, so that the network remains functionally unchanged.
>
> [1] Wang, Dequan, et al. "Tent: Fully test-time adaptation by entropy minimization." arXiv preprint arXiv:2006.10726 (2020).
> [2] Ainsworth, Samuel K., Jonathan Hayase, and Siddhartha Srinivasa. "Git re-basin: Merging models modulo permutation symmetries." arXiv preprint arXiv:2209.04836 (2022).

---

> > ### Comment · Reviewer_zoWW · 2022-11-28
> > **Raised my score to 6**
> >
> > Dear Authors,
> >
> > Thank you for the detailed response. My main concerns are addressed. Hence, I am raising my score to 6.

---

### Official Review · Reviewer_LkzC · 2022-10-29

**Confidence:** 3
**Correctness:** 3
**Technical Novelty And Significance:** 3
**Empirical Novelty And Significance:** 3
**Recommendation:** 8

**Clarity, Quality, Novelty And Reproducibility:**

The paper is well-written and refreshingly honest about the LMC phenomenon being shown empirically for smaller models while still needing further validation in the ImageNet setting where the energy barrier is still significant. The overall idea is simple and straightforward.

The authors do provide source code, so reproducibility is likely.

**Strength And Weaknesses:**

Strengths:
- Renormalization without permutation significantly lowering the barrier is an interesting finding complementary to the concurrent Ainsworth et al. 2022 result.
- Renormalization technique is an improvement over Ainsworth et al. 2022, which requires the replacement of BatchNorm layers with LayerNorm layers to alleviate the barrier in Residual networks.

Weaknesses:
- The pattern of increasing test error corresponding to a larger barrier is an interesting finding. But there doesn't seem to be an intuition on why this is true. Without an explanation, it's unclear why this pattern is viewed as expected or "simple"

From the paper:
"the high barrier of ImageNet models stems in part simply from the relatively higher test error of said models"

Comment:
Could the authors elaborate on the connections between robustness and linear mode connectivity? Is there an expectation that linear mode connectivity will indicate robustness?

**Summary Of The Paper:**

This work proposes the striking empirical phenomenon that, accounting for permutations, two independently trained networks of the same architecture will have no energy barrier on the linear path between them. Furthermore, the authors demonstrate an issue with naive interpolation for deeper networks where the activations experience a variance collapse on this linear path, creating an 'artificial' barrier on this connected path. To resolve this, the authors propose a simple renormalization procedure that re-establishes mode connectivity (albeit no longer linear).

**Summary Of The Review:**

The empirical validation of the LMC phenomenon is important, and this work does a good job of diagnosing the issues with LMC when applied in practice. The renormalization method appears novel and significantly improves the loss barrier along the bath between the two models.

---

> ### Author Response · Authors · 2022-11-15
> **Response to Reviewer LkzC**
>
> We thank the reviewer for their recognition of the effectiveness of our method over the state-of-the-art and its value in the context of barrier reduction.
>
>     Relation between the test error and the barrier size
> We support the notion that easier tasks have naturally lower barrier to interpolation. Entezari et al (their Figure 4.c) observe that simpler tasks have lower barrier before permutation. Similarly we observe a positive relationship between endpoint test error and barrier size after applying REPAIR across our experiments on CIFAR-10. Also similar related works such as Pittorino et al. have observed optimization algorithms such as RSGD which finds flatter solutions (and lower test error) contribute to lower barrier (their Figure 9). Therefore, we hypothesize that models that get better test accuracy on ImageNet would result in lower barrier (before alignment and after alignment + REPAIR).
>
> [1] Pittorino, Fabrizio, et al. "Deep Networks on Toroids: Removing Symmetries Reveals the Structure of Flat Regions in the Landscape Geometry." arXiv preprint arXiv:2202.03038 (2022).
>
>     Robustness and linear mode connectivity
> There are numerous works that show flatter minima help generalization (adversarial robustness, OOD robustness). [1] suggested “sharpness” (robustness of the training error to perturbations in the parameters) as a complexity measure for neural networks. [2] show that worst-case flatness correlates well with better generalization, e.g., for small batch sizes.  [3] argues that normalization is helping  generalization by allowing to find flatter minima. On the other hand, flatter minima increase connectivity of linear interpolation between solutions [4].
>
> [1] Keskar, Nitish Shirish, et al. "On large-batch training for deep learning: Generalization gap and sharp minima." arXiv preprint arXiv:1609.04836 (2016).
>
> [2] Jiang, Yiding, et al. "Fantastic generalization measures and where to find them." arXiv preprint arXiv:1912.02178 (2019).
>
> [3] Bjorck, Nils, et al. "Understanding batch normalization." Advances in neural information processing systems 31 (2018).
>
> [4] Frankle, Jonathan, et al. "Linear mode connectivity and the lottery ticket hypothesis." International Conference on Machine Learning. PMLR, 2020.

---

### Comment · Area_Chair_XfCz · 2022-11-19
**Question**

Dear Authors,

I'm curious, given the rescaling method, how important is the neural alignment?
In other words, I only see experiments for:
1. interpolation
2. neural alignment + interpolation
3. neural alignment + rescaling + interpolation

But what happens if we do
4. rescaling + interpolation ?

---

> ### Author Response · Authors · 2022-11-21
> **It depends on the interpolation coefficient**
>
> Dear Area Chair,
>
> We found this an intriguing question, and have added several such experiments as a new figure.
>
> New paper uploads are no longer allowed, so here is an anonymized repository containing the figure: https://anonymous.4open.science/r/REPAIR-nopermutation-5010/README.md
>
> It appears that towards the endpoints of the interpolation (that is, with α ≤ 0.2 or α ≥ 0.8), the neural alignment makes little difference, but the rescaling is essential. On the other hand, towards the midpoint the neural alignment becomes crucial for high performance.
>
> Best regards,
> The Authors

---

### Decision · Program_Chairs · 2023-01-20

**Decision:**

Accept: poster

**Justification For Why Not Higher Score:**

It's a simple and nice method, but in retrospect, it not is very surprising that it works.

**Justification For Why Not Lower Score:**

All reviewers vote to accept [6,6,8].

**Metareview: Summary, Strengths And Weaknesses:**

This paper examines neural networks which are interpolations between two SGD solutions, and finds that, even after alleviating issues related to permutations (using existing algorithms), we get high loss barriers due to "variance collapse", i.e., a collapse in the variance of their activations, causing poor performance. An algorithm is suggested to repair this issue by rescaling the reactivations, which helps a lot with the final performance.

The reviewers had some concerns, and this was initially a borderline paper. However, most concerns were addressed after a major revision of the paper (almost being a "too large" revision of the paper). The only remaining possible weakness seems to be the novelty of the method: it is very simple, and reminiscent of BN-tuning which was previously suggested (note to authors: previous papers already did this even without backpropagation [A], and without the pre-existence of a BN layer [B]). However, the specific method of initialization of the BN layers differs from previous papers. Anyway, I don't think this is much of an issue, as I think the main novelty here is that such a simple method is so useful for this different problem.

[A] Sun et al., Hybrid 8-bit Floating Point (HFP8) Training and Inference for Deep Neural Networks, NeurIPS 2019

[B] Hubara et al., Accurate post training quantization with small calibration sets, ICML 2021

**Note From Pc:**

if the above contains the word "oral" or "spotlight" please see: "oral" presentation means -> notable-top-5% and "spotlight" means -> notable-top-25%. As stated in our emails, we are disassociating presentation type from AC recommendations